# Recent Advances in Biomass-Derived Carbon Materials for Sodium-Ion Energy Storage Devices

**DOI:** 10.3390/nano12060930

**Published:** 2022-03-11

**Authors:** Mengdan Yan, Yuchen Qin, Lixia Wang, Meirong Song, Dandan Han, Qiu Jin, Shiju Zhao, Miaomiao Zhao, Zhou Li, Xinyang Wang, Lei Meng, Xiaopeng Wang

**Affiliations:** College of Science, Henan Agricultural University, Zhengzhou 450001, China; yan024711@163.com (M.Y.); wanglixia@henau.edu.cn (L.W.); smr770505@henau.edu.cn (M.S.); handd@henau.edu.cn (D.H.); jinqiukl@henau.edu.cn (Q.J.); zhaoshiju008@henau.edu.cn (S.Z.); zmm2581472022@163.com (M.Z.); lizhou_1995@163.com (Z.L.); wxy1656135@163.com (X.W.)

**Keywords:** biomass-derived carbon, energy storage, sodium-ion battery, sodium-ion capacitor

## Abstract

Compared with currently prevailing Li-ion technologies, sodium-ion energy storage devices play a supremely important role in grid-scale storage due to the advantages of rich abundance and low cost of sodium resources. As one of the crucial components of the sodium-ion battery and sodium-ion capacitor, electrode materials based on biomass-derived carbons have attracted enormous attention in the past few years owing to their excellent performance, inherent structural advantages, cost-effectiveness, renewability, etc. Here, a systematic summary of recent progress on various biomass-derived carbons used for sodium-ion energy storage (e.g., sodium-ion storage principle, the classification of bio-microstructure) is presented. Current research on the design principles of the structure and composition of biomass-derived carbons for improving sodium-ion storage will be highlighted. The prospects and challenges related to this will also be discussed. This review attempts to present a comprehensive account of the recent progress and design principle of biomass-derived carbons as sodium-ion storage materials and provide guidance in future rational tailoring of biomass-derived carbons.

## 1. Introduction

In recent years, the increasing demand for renewable and cleaner energy re-sources such as wind, solar, and wave, to replace traditional fossil energy, has required the development of cost-effective, high-performance, large-scale energy-storage systems. Lithium-ion batteries (LIBs) with the advantages of high energy density have been widely used in the field of energy-storage systems. However, global lithium resources are limited and unevenly distributed, which will make the cost of LIBs dramatically increase shortly. Therefore, low-cost energy-storage systems using naturally abundant raw materials have attracted extensive attention. Na-ion energy storage devices (SESDs), including sodium-ion batteries (SIBs) and sodium ion capacitors (SICs), are recognized as alternatives to LIBs due to the high overall abundance of precursors and better cycle stability and power density [1,2].

The sodium resource is rich in terms of reserves (2.74% of the earth’s crust) and ranks fourth among metal elements, has an even geographical distribution, and has a distinctly lower cost than lithium. Sodium and lithium, located in the same main group, have similar physical and chemical properties [3]. Therefore, sodium has attracted increasing attention as a potential alternative to lithium in electrochemical energy-storage systems, especially for grid storage [4,5,6,7,8,9]. Many fundamental understandings of LIBs and Li-ion capacitors (LICs) provide much experience for the research of sodium-ion energy storage devices. Although sodium and lithium are chemically similar, they still have some differences. For example, Na^+^ ions are larger and heavier than Li^+^ ions (ion radius 1.02 Å vs. 0.76 Å, weight 23 g mol^−1^ vs. 7 g mol^−1^) [2,10,11,12], resulting in worse diffusion kinetics in most host materials and inferior gravimetric/volumetric capacity. The redox potential of Na^+^/Na is 0.3 V above that of Li^+^/Li, which reduced operating voltage and energy density [13,14]. Thus, promising electrode materials with suitable working voltage and sodium storage performance can be one of the potential solutions to make up for the loss. In previous studies, great progress was made in the exploration of cathode materials. A large variety of cathode materials (e.g., oxides and polyanionic compounds) can effectively store sodium ions [15,16], although there is the problem of the bottleneck at the anode. Present anode materials used for sodium-ion storage mainly include alloys, metal oxides/sulphides, organic compounds, titanium-based materials, and carbonaceous materials [1]. Among many anode materials, carbon-based materials offer multiple advantages such as higher capacity, lower average sodium storage potential, good conductivity, being non-toxic, and having a low price and hence are recognized as competitive candidates for SESDs [17,18].

Unlike lithium-ion storage, graphite is unfavorable for hosting sodium-ion in graphene layers [19,20]. Various types of amorphous carbon, including soft carbon and hard carbon, have been examined that can be used as potential anode materials for SESDs [19,21,22,23]. Amorphous carbons with a large interlayer spacing of 0.36–0.4 nm make it a suitable Na hosting. However, soft carbon generally exhibits low initial coulombic efficiency (ICE) and a specific capacity. Hard carbon contains random stacked graphitic layers and a tortuous structure, which allow it to effectively store Na-ion in the micropores and exhibit high reversible capacity and good kinetic performance [24,25,26]. Owing to the renewability, low cost, and diverse inherent structure of biomass, its use as a precursor for producing amorphous carbon to reduce costs and improve electrochemical performance has become an important research direction to satisfy the requirements for its practical application in the SESDs field.

Biomass refers to widely distributed living and growing organic materials, such as plants [27,28,29], microorganisms, and animals [30], that have been pyrolyzed at high temperatures to become biomass-derived carbon materials. Reasonable utilization of biomass will realize “turning waste into treasure”. Most importantly, biomasses have their unique microstructures and compositions, and the resultant biomass-derived carbons usually retain the diversity of their structures and compositions after pyrolysis (Figure 1) [31,32,33,34,35]. Different structures (e.g., hard carbons, soft carbons, and hybrid carbons), different compositions (e.g., N-doped carbons and other atom doped carbons), and different morphologies (e.g., 1D, 2D and 3D hierarchical structures) of biomass-derived carbons greatly affect their electrochemical performance in SESDs. Table 1 shows the application potential of different types of carbon materials in the field of Na-ion storage. The various morphology, structure, and electrochemical performances of carbonaceous materials have been compared [36,37,38,39,40,41,42,43,44,45,46,47,48,49,50,51,52,53,54,55,56]. The influence of different types of biomass-based carbon materials on the electrochemical performance of SESDs should be further systematically summarized. The understanding of the biomass-derived carbons and their storage mechanism can be reviewed to guide a rational design for effective electrode materials for SESDs.

Herein, we attempt to provide a comprehensive summary of the latest developments of various biomass-derived carbons used in SESDs, including the principle of sodium ion storage in SIBs and SICs, and the classification of biomass carbon with different structures and compositions. The recent progress and electrochemical performance of different types of biomass-derived carbons will be introduced in detail. This review focuses on the influence of different micromorphology and compositions of biomass-derived carbon on electrochemical performance. Finally, the challenges and perspectives for SESDs have also been proposed. We hope that this literature review can provide references for the rational design of carbon materials toward high-performance SESDs.

## 2. Sodium-Ion Storage Mechanism in Carbonaceous Materials for SESDs

### 2.1. Configuration and Mechanism of Sodium-Ion Batteries

Sodium-ion batteries have come back into the spotlight, due to their potential cost advantages, since 2010. SIBs consist of cathode materials and anode materials separated by the electrolyte. The energy storage of SIBs is realized through sodium ions shuttling between cathode and anode materials in the charge/discharge process. During the charging process, sodium ions are extracted from cathodes and accommodated into anodes transport through the electrolyte. The reverse reaction occurs in the discharge process. The widely-accepted theory suggests that SIBs operate on a similar intercalation mechanism to LIBs. However, sodium ion storage mechanism in anode materials is more complex than that of lithium insertion in graphite. Graphite is usually not suitable for sodium intercalation in traditional ester-based electrolytes, which leads to unsatisfactory capacity and cyclability of SIBs [57]. Thus, many new materials such as carbonaceous materials, alloys, metal oxides/sulfides, and titanium-based compounds have received widespread attention as anodes for SIBs [58,59,60,61,62]. Amorphous carbons, especially hard carbons, exhibited greatly improved performance and industrial feasibility for SIBs. They are competitive enough for practical application in large-scale grids, compared with other non-carbonaceous materials.

The current carbonaceous materials that have been widely investigated for SIBs mainly include graphite-based carbon materials, soft-carbons, and hard-carbons (Figure 2a–c). Theoretically, it is difficult to intercalate Na ions into graphite interlayer (Figure 2d). The formation energies of sodium-graphite intercalation compounds are not stable [63]. However, taking advantage of the co-intercalation effect of solvated Na ions, natural graphite in some linear ether-based electrolytes exhibited unexpected rate capability and cyclability (100 mAh g^−1^ at 10 A g^−1^, and 95% capacity retention after 6000 cycles) [64]. The mechanism of Na-ions storage has been proposed so that solvated ions might be adsorbed in graphite lattice rather than atomically bonded to carbon. The expanded graphite with enlarged interlayer distance showed reversible capacity of 284 mA g^−1^ [36]. The experiment confirmed the efficient Na-ions insertion/extraction mechanism in lattice of expanded graphite, which is different from other carbonaceous materials. Reduced graphene oxide has been studied as the anode materials showed a reversible capacity ~450 mAh g^−1^ at a current density of 25 mA g^−1^ [65]. An adsorption mechanism of graphene has been demonstrated for Na-ions storage [56,66]. The structure engineering of graphene to control the defect and surface area is an effective strategy for Na-ions storage.

Soft carbon is non-graphitic carbon that can be graphitized at high temperatures, whose graphitizable degrees and interlayer distance can be tuned by thermal treatment. Soft carbon, similar to graphite, is always smooth and exhibits less curvature in the graphitic layer. The Na-ion storage capacity of soft carbon was first demonstrated by Doeff and colleagues to show about 90 mAh g^−1^. The improved capacities and good rate performance have been reported by many groups. For instance, using anthracite and aromatic compounds as precursors, the prepared soft carbons showed high reversible capacities above 200 mAh g^−1^ [43]. However, the Na-storage mechanism of soft carbon revealed that Na-ions adsorption on isolated graphene sheets of soft carbon contributes to the general sloping potential profile and the lack of low-potential plateau, which could cause negative effects on energy density and potential safety of SIBs.

Compared with soft carbon, hard carbon has received more attention as anode in SIBs due to its good kinetic performance [24,46,71]. Hard carbon is non-graphitizable carbon that has curved and unaligned graphitic layers. The crosslink graphitic layers and disordered structures are generally considered to be favorable for Na-storage performance. The typical charge-discharge curves of hard carbon include high-potential sloping region and low-potential plateau region (Figure 2e). These potential regions indicate the reaction of Na-ions with different structures. In 2000, Stevens and Dahn [72] underly the mechanism of Na-ions insertion into hard carbon, suggesting that the high-potential sloping region corresponded to the insertion of Na ions inside graphitic layers and adsorption on the defect sites, whereas Na ions were inserted into nanopores along the low-potential plateau. The alternative storage mechanism is also proposed by Cao and colleagues [69]. They conclude that Na-ions are stored at defect sites of the surface in the sloping part of the potential curve and inserted into graphitic layers and pores in the low-potential plateau region. Based on the understanding of existing sodium storage mechanisms, rational, structural, and defect engineering will be effective strategies for the design of hard carbon with higher performance.

### 2.2. Configuration and Mechanism of Sodium-Ion Capacitors

Among the sodium-ions based energy storage devices, SIBs and SICs are currently prominent. The battery utilizes intercalation mechanism, leading to high energy density and limited power density. Supercapacitor with adsorption-desorption mechanism provides high power density but limited energy density. SICs have attracted much attention due to their combined advantages of battery and supercapacitor [73,74]. A typical SIC is generally composed of a capacitive cathode and battery-type anode and is the most commonly used type. On the contrary, other configurations of SICs include a battery-type cathode and capacitive anode. The configuration of SICs determines the charge-storage mechanism. This section will focus on the charge-storage behavior of carbonaceous materials in the first widely used configuration.

Dual-carbon SICs were first reported by Kuratanni and colleagues [75]. This SIC includes a battery-type anode of hard carbon and a capacitive cathode of activated carbon. In the case of dual-carbon SICs, the capacitive cathode of activated carbon stores charges through a non-faradaic surface ion adsorption mechanism on the interface of electrode and electrolyte. The faradaic reaction on the surface or near-surface of battery-type anodes, such as hard carbon, provided a higher capacity for SICs. In general, the two kinds of energy storage mechanisms, including electrochemical double-layer capacitance (EDLC) and pseudocapacitive behavior, are responsible for electrochemical reactions of SICs. The EDLC operates on the mechanism of electrolyte ions’ adsorption/desorption on the surface of the electrodes. In the case of pseudocapacitive behavior, charge storage mainly originates from the electron-transfer rather than the adsorption of ions. Different from the storage mechanism of SIBs, when carbonaceous materials are used as anode in SICs, the faradaic redox reaction occurs only on the surface of the electrode, and pseudocapacitive intercalation does not produce phase transition. Therefore, SICs based on this hybrid mechanism provide an effective route for integrating high energy and power performance.

## 3. Diverse Morphology of Biomass-Derived Carbons for SESDs

An in-depth understanding of sodium-ion storage mechanisms in SESDs, especially using carbon-based materials as electrodes, can greatly improve the rational design of better electrode materials. Exciting theoretical studies suggest that morphological and composition engineering (such as defect sites, nano-porosity, and heteroatom doping) are the potential strategies to develop effective electrode materials. Biomass-derived carbons have attracted much attention for application in SESDs in recent years. Different treatment methods and precursor materials will achieve different carbonaceous materials with inherent macroscopic morphologies and structures, resulting in a variety of electrochemical behaviors [76,77]. This section mainly introduces the recent advances of diverse morphologies and structures of biomass-derived carbons used for SESDs. The relationship between the structure/morphology and electrochemical performance of different biomass-derived carbons will be discussed in detail.

Biomass materials naturally possess abundant and diverse macrostructures ranging from zero to three dimensions that can be inherited and evolved by corresponding biomass-derived carbon materials. So far, various nanostructured carbon materials with zero-dimensional (0D) spherical structures, one-dimensional (1D) nanofibers/nanotubes, two-dimensional (2D) nanosheets, and three-dimensional (3D) hierarchical structures have been synthesized. The main motivations to create electrode materials with different dimensional nanostructures are to enlarge exposed active surface areas, broaden activated ionic channels, and accelerate electron conductivity, all of which can significantly promote the electrochemical performance of SESDs. The recent progress and simple classification of biomass-derived carbons based on dimensions are shown below.

### 3.1. Tubular and Fiber-Shaped Biomass-Derived Carbons

As 1D nanostructures, tubular and fiber-shaped carbon precursors are widely distributed in nature, such as plant tissues [45,78] and bacterial secretions [53]. 1D carbon materials have high aspect ratio to provide fast channels for electron and ion transport. Especially, the tubular structure of carbon also forms an effective permeable inner surface structure that is conducive to Na^+^ adsorption, shortening ion diffusion. The path accelerates the diffusion and migration of electrolyte ions from the electrolyte to the inside of the surface of the electrode during charge and discharge [46]. Carbon materials with tubular structures are considered as viable structures for high-performance sodium-ion energy-storage applications.

Li and co-workers [46] reported the preparation of hard carbon materials that maintain uniform microtubule shapes using renewable natural cotton biomass as a precursor (Figure 3a,b). The hollow tubular structure of the hard carbon material is beneficial to the migration of the electrolyte, reduces the diffusion distance of Na^+^ ions, and improves the electrochemical performance of the hard carbon. Yu et al. [47] prepared the carbon with micro/nanotubular structure from low-cost *kapok* fibers. During the carbonization process, the *kapok* fibers underwent aromatization, polycondensation, and the formation of short graphite layers, maintaining good morphology with a highly specific surface area. The gradual reduction of carbon micro-nanotubes with the increase of carbonization temperature is expected to reduce the formation of SEI; improve the initial Coulombic efficiency; and finally yield carbon micro-nanotube samples with high reversible capacity, high initial Coulombic efficiency, and excellent rate performance at a carbonization temperature of 1400 °C. Tubular biomass carbon is considered one of the most promising anode candidates for sodium-ion batteries (SIBs) due to its abundant natural resources, low cost, and sustainability, to prepare high-performance sodium storage media with excellent microstructure and morphology. Liu and co-workers [79] proposed to prepare interconnected porous carbon frameworks that maintain a good tubular hierarchical porous structure through KOH activation and Co^2+^-assisted graphitization (Figure 3c). The interconnected macropores improve the mass transfer between the electrolyte and the active material and shorten the diffusion distance of Na^+^ to the inner surface of the carbon framework and improve the excellent electrochemical performance of carbonaceous materials. The above studies provided a reference for fully exploiting the advantages of the inherent 1D morphology of biomass precursors and highlighted the importance of tubular structures in sodium-ion storage.

### 3.2. Sheet-Shaped, Biomass-Derived Carbons

Different precursors and pretreatment methods often have a serious impact on the properties of hard carbons (Figure 4). Using cucumber stems as precursor [51], the carbonization temperature was adjusted to 1000 °C to prepare sheet-like hard carbon anode materials. A reversible capacity of 337.9 mAh g^−1^ and Coulombic efficiency of 99–100% after 500 cycles were obtained. The coiled hard carbon materials were successfully prepared by a two-step method of hydrothermal treatment and pyrolysis at different temperatures using biomass templates. Flake-like hard carbon extracted from pistachio shell precursors were prepared with coiled hard carbon materials at different temperatures [80]. The anode provided a high capacity of 317 mAh g^−1^ with larger interlayer spacing when carbonized at 1000 °C. It showed that carbonization temperature and morphology control have a great influence on the electrochemical performance of hard carbon materials. A hard carbon nanosheet anode made of cherry petal [81] precursor was synthesized at 1000 °C. At a current density of 20 mA g^−1^, its relatively stable capacity is 300.2 mAh g^−1^ and the initial Coulombic efficiency is 67.3%. The synergistic effects of the mesoporous structure and the increased interlayer distance during the pyrolysis of the precursor of *cherry petals* enhanced the storage capacity of sodium ions. In addition, sheet-like structured hard carbon anode materials were prepared using *oat flakes* [82], biomass-based gelatin [83], and *maple* [84], as precursors to study their sodium storage properties.

Using oat flakes as the precursor [82], two-dimensional hard carbon was obtained by carbonization (Figure 4a). According to the BET in Figure 4b, the hard carbon contained a large number of mesopores. Compared with other carbon materials with highly specific surface area, the formation of SEI film is limited, thereby improving the Coulombic efficiency. Jin synthesized N, B co-doped carbon nanosheets [83] using biomass-based gelatin as the precursor and boronic acid as the template (Figure 4c). The synergistic effect of heteroatom doping and 2D structure with highly specific surface area enhances the capacity and rate performance of Na-ion batteries. Wang et al. [84] developed a hard carbon extracted from maple tree as the anode of the battery, as shown in Figure 4d, which achieved a capacity of 337 mAh g^−1^ at 0.1 C. The initial Coulombic efficiency was as high as 88.0%. The capacity remained at 92.3% after 100 cycles at 0.5 C. Again, it is proved that biomass-derived hard carbon has the advantages of large capacity and high Coulombic efficiency. The hard carbon materials showed interlayer spacing suitable for Na ion insertion, with highly defective sites and specific surface area, which can effectively improve the weight/volume capacity and superior cycling stability.

### 3.3. 3D Hierarchical Structures of Biomass-Derived Carbon

3D carbons have highly interconnected network, abundant active edges, defects, shortened ion/electron channels, accelerated dynamic ion transfer, and good electrical contacts, thus generally possessing excellent electrochemical performance [85,86]. The 3D structure can not only provide a continuous electron path but also allows the electrolyte to penetrate the whole structure and facilitate the sodium ion transport between the electrode/electrolyte interface by shortening the diffusion path to ensure good electrical contact to facilitate ion transport [87]. However, due to the natural hydrophobicity of the carbon matrix surface, its effective specific surface area is greatly limited by the highly developed single microporous structure. Therefore, improving the accessible specific surface area is an effective way to improve the electrochemical performance.

Hierarchical porous structures including macropores, mesopores, and micropores can increase the contact area of the electrolyte with the electrode material [48,88]. Among them, macropores (pore size > 50 nm) act as ion buffer reservoirs during the charging and discharging process, which is conducive to the transport of substances and the proximity of ions to adsorption sites, effectively shortening the diffusion path of electrolyte ions and ensuring the rapid transmission and diffusion of electrolyte ions. The mesopores (50 nm > pore size > 2 nm) increase the contact area between the electrode and the electrolyte, which can act as a reservoir for the electrolyte, reducing the ion diffusion resistance and speeding up the ion transfer pathway. Micropores (pore size < 2 nm) can generate a higher surface area, provide a large number of active sites, and thus increase the specific capacitance. This unique hierarchical pore structure carbon with a highly specific surface area provides an efficient route for the penetration and transport of electrolyte ions and is expected to be an excellent anode material for Na-ion energy-storage systems.

Zhang synthesized a unique “honeycomb” structure carbon (Figure 5a) that used pine pollen as a precursor [89]. Their hollow structure and robust framework reduced volumetric strain during Na-ions intercalation/deintercalation and rapidly accommodated ions/electrons for better rate performance. The initial discharge capacity can reach 370 mA h g^−1^ at a current density of 0.1 Ag^−1^. After cycling 200 times, the reversible capacity also stabilized at 203.3 mA h g^−1^ with a retention rate of 98%. The high capacity and long lifetime of SIBs mainly benefit from the biomimetic honeycomb structure and robust carbon framework. These properties can accelerate ion transport, shorten charge diffusion paths, and effectively buffer volume expansion. In addition, the carbon film prepared by carbonization of *Osmanthus fragrans* leaves provided a conductive framework and provided better nucleation conditions for the in-situ growth of transition metal phosphides. The Fe-doped CoP has a flower-like structure composed of intersecting nanoflakes of 200–300 nm (Figure 5b). Due to the large surface area of the flower shape, which provided more active sites for the intercalation of Na ions and the strong coupling between Fe-doped CoP and the carbon film, the Na ion storage performance was significantly improved. During the electrochemical reaction, the carbon film with high conductivity was beneficial to electron transfer. Fe-doped CoP/C maintained a specific capacity of 324 mAh g^−1^ after 500 cycles, with a Coulombic efficiency of about 99%.

Lang and co-workers [52] constructed a novel sodium-ion hybrid battery (SHB) by introducing adsorption-type hierarchical porous amorphous carbon (HPAC) as the anode material. SHB has good rate performance and long-cycle cycling performance at 2C, and the capacity retention is 87% after 1000 cycles at 10C. Qin et al. [90] studied the pore size distribution of wheat straw after carbonization at different temperatures. At 900 °C, the pore size distribution was mainly about 3.8 nm, and the specific surface area was 1295.21 m^2^ g^−1^. Pore-rich biochar facilitates electrolyte diffusion and Na ion transport and can expand the interlayer spacing of graphite to de/intercalate Na ions, improving battery performance with higher stable reversible capacity. In addition, Luo and colleagues [88] prepared a novel hierarchically structured porous carbon material (Figure 5c–e) with macropores, mesopores, and micropores by activating longan shells. The highly specific surface area and excellent porous structure ensure its good sodium-ion storage and cycling performance.

Biochar has a variety of microstructures, and different microstructures largely affect the electrochemical properties of electrochemically active sites and surfaces [91]. The performance of Na-ion batteries can be tuned by increasing the specific surface area by controlling the microstructure of biomass carbon [30,68,92,93,94]. Hu et al. [95] prepared carbon nanosheets with pinecone shells as a precursor. Under the synergistic effect of KOH and melamine, discrete carbon nanosheets with large specific surface area and rich porosity can be prepared. This structure ensures its excellent energy storage, showing excellent rate performance and excellent cycle performance. A porous tubular carbon material was synthesized by using sycamore single villi as a precursor [96]. This electrode material had a specific capacitance of 836.4 F g^−1^ at a current density of 0.2 A g^−1^ and retained a specific capacitance of 92.96% after 10,000 cycles at a current density of 10 A g^−1^. Wang et al. prepared a hierarchical porous carbon material containing a large number of micropores and a small number of mesopores using Paulownia husk as a raw material. The obtained biomass char has a surface area of 1914.4 m^2^ g^−1^. The porous structure exhibits excellent rate capability, with a discharge capacitance of 100 mAh g^−1^ at a current density of 1 A g^−1^ after 100 cycles. Junke Ou and co-workers [97] used human hair as raw material to prepare nitrogen-doped porous carbon, which can provide a high capacity of 308 mAhg^−1^ at a current density of 100 mAg^−1^. In conclusion, the unique hierarchical microstructure increases the electrode-electrolyte contact area, modulates the volume expansion during cycling, and significantly improves the electrochemical performance.

## 4. Different Structures and Components of Biomass-Derived Carbons for SESDs 

### 4.1. Degree of Graphitization

Carbon materials are mainly divided into graphite, graphene, soft carbon, and hard carbon. Graphite is electrochemically less active for Na storage due to thermodynamic problems. To further improve electrochemical sodium-storage properties, graphene was applied as the anode material and performed better than graphite. Nitrogen-doped 3D graphene foams have been prepared to deliver a high initial reversible capacity of 852.6 mAh g^−1^ at 1 C [98]. However, the low initial Coulombic efficiency (~18.5%) of graphene owing to the irreversible Na_2_O formation on graphene surface limits its practical application [46]. Non-graphitic carbon materials, including soft carbon and hard carbon, have been widely used as anode materials for SIBs. However, the capacity of soft carbon is lower than that of hard carbon. Among the many carbon materials, hard carbon has attracted extensive attention. Amorphous regions in hard carbon materials are often embedded in graphite layers, forming a strong cross-linked network that makes the structure more rigid. With pores between randomly arranged graphite crystallites, the structure affects storage sites, and diffusion kinetics. So, the electrochemical performance can be changed by the degree of graphitization [99,100,101]. Many studies have reported that with the increase of carbonization temperature [28,102,103,104,105,106], the degree of graphitization will increase, the defects of hard carbon will decrease, and the structure will gradually become ordered.

In 2019, Stevanus [105] carbonized fir wood under different high-temperature conditions in Figure 6. With the increase of carbonization temperature, the hard carbon gradually became ordered from high defects, and the spacing became smaller, which was not conducive to the insertion of Na^+^ and was not thermodynamically stable. In addition, biomass-derived carbon contains randomly arranged graphite layers and disordered layered nanodomains, which cannot be fully graphitized even at temperatures above 3000 °C. Even if the carbonization temperature is increased, the material will have defects, but the number of defects will decrease. Cao [93] prepared rapeseed into layered hard carbon. When the spacing is 0.39 nm, it can ensure the insertion and extraction of Na^+^ when it is used as a negative electrode material for SIBs, so it has excellent electrochemical performance. With the increase of temperature, the capacity increases first, because the carbon particles are gradually connected tightly, which is conducive to the transport of electrons. However, the capacity starts to drop after 700 °C, mainly because of the formation of stacked blocks, which hinder the electron transport. Although the partial carbonization of hard carbon can effectively improve the reversible capacity of Na ion intercalation, an overly high temperature will further reduce the interlayer spacing, reduce the pore volume, and cannot accommodate Na ion insertion or adsorption to the pore surface, resulting in a decrease in capacity [107,108].

### 4.2. Heteroatom Doping

In order to improve the Na storage of carbon materials to meet the needs of energy storage in various aspects, using carbon materials doped with heteroatoms (N, S, P, B, O, etc.) is an effective strategy [48,88,92,109]. Heteroatom doping usually can improve conductivity, increase active sites, and expand interlayer spacing. The intercalation/deintercalation of sodium ions in the electrochemical process is promoted, and the reversible capacity of bio-based carbon is several times larger than the theoretical capacity of graphite [110].

Nitrogen-rich doped carbon spheres were synthesized using onion waste as the precursor [111]. Nitrogen doping enhances the extension of the interlayer distance, which is favorable for the insertion/extraction of large Na ions (Figure 7a,b). The considerable amorphous structure and heteroatom doping enhance the electrical conductivity and active sites of the material; the reversible capacity is also enhanced, and the structural deformation during cycling is alleviated. In addition, the synergistic effect of binary/multiple heteroatoms can not only obtain larger interlayer spacing and provide additional charge storage capacity by Na^+^ binding to relevant defects or functional groups, it can also contribute to the conduction band of carbon by providing additional free electrons, resulting in higher electrical conductivity and improved electrochemical performance [92,97,112,113]. Jin synthesized N, B-doped carbon nanosheets by a one-step carbonization method using biomass gelatin as the precursor and boronic acid as the template [83]. The addition of N, B will produce more defects and disordered structures. The differential charge density and density of states are calculated by building a heteroatom doping model, indicating superior electrochemical performance. In addition, Liu et al. [5] synthesized a N, O co-doped porous carbon with uniform ultra-micropores. The presence of N atoms helps to improve the electrical conductivity, while the oxygen functional group can improve the wettability of the electrode material and promote better contact between the active material and the electrolyte ions to improve the electrochemical performance (Figure 7c–h). Overall, heteroatom-doped carbon materials are considered promising anode candidates for SESDs.

### 4.3. Hybridization of Biomass-Derived Carbon and Metal Compounds

Transition metal oxides, sulfides, and phosphides have high theoretical capacities [114]. However, as electrode materials for sodium-ion energy-storage systems, the volume changes during the charge and discharge process, and the electrodes are severely pulverized, resulting in low energy storage density and poor cycle performance. To improve these problems, the most effective method is to hybridize transition metal oxides, sulfides, or phosphides with carbon materials [53,54,115,116]. This not only provides a conductive network for electron transfer but also acts as a stable structural matrix to accommodate volume changes during cycling. Among carbon materials, biomass carbon has high thermal/chemical stability, unique morphological structure, and high electrical conductivity, especially biomass containing different functional groups, such as hydroxyl and amino. These functional groups are easily combined with metals, so biomass carbon becomes the best candidate for metal composites.

In 2016, Yang and his team first reported the double-helix three-dimensional metal sulfide/carbon aerogel nanostructures combined with carrageenan-metal hydrogel for high-performance sodium-ion storage (Figure 8a). Using it as an electrode material, it showed a high reversible specific capacity of 280 mAh g^−1,^ even after 200 cycles at a current density of 0.5 Ag^−1^ [117]. The carbon skeleton in this nanostructure not only facilitates the fast charge transfer reaction but also enhances the mechanical properties of FeS nanoparticles and buffers their volume changes, thereby extending the electrode cycle life. Moreover, Ni_3_S_4_ nanoparticles were embedded in porous carbon (Figure 8b,c) [118]. As a negative electrode for Na-ion batteries, it maintained a capacity of 297 mAh g^−1^ for 100 cycles at a current density of 1 A g^−1^. Its excellent electrochemical performance benefits from porous carbon inhibit the accumulation of Ni_3_S_4_ nanoparticles during the synthesis. In addition, the addition of Ni_3_S_4_nanocrystals accelerates the transport of sodium ions, thereby improving the capacity and reaction kinetics. In conclusion, metals are intercalated into biomass-derived carbon as active materials, providing more active sites, while biomass carbon limits the volume change during the intercalation/deintercalation of sodium ions through internal stress. The synergy between the two together improves the stability and electrochemical performance of Na-ion batteries [32,114,118,119,120].

## 5. Conclusions, Challenges, and Outlook

Different biomass carbon materials with their inherent structure and chemical advantages have opened up a new key field for the design and preparation of electrodes for SESDs. Due to its wide range of sources, non-toxicity, and chemical stability, the application potential of biomass carbon materials in sodium-ion energy-storage systems is believed to help meet future environmental needs. This article reviews the latest developments in the application of sodium-ion batteries and sodium-ion capacitors with biochar materials of various structures, morphologies, and chemical compositions, and the factors that affect electrochemical performance. This provides references for the future tailoring of advanced carbon materials for SESDs.

Although biomass carbon electrodes have great potential in SIBs and SICs, there are still some problems that need to be solved before they can be successfully commercialized and widely used. Their further challenges are mainly as follows. 1. Biomass carbon source materials are difficult to use to achieve high-quality, uniform mass production, owing to the diversity of geography and environment. 2. The relatively low carbon yield of precursors limits its industrialization [37,38,39,41,50]. For this case, industrial products derived from biomass can be utilized for synthesizing uniform electrode materials for SESDs. This will be one of the feasible methods for practical production. 3. The impurities in biomass carbon are usually detrimental to the electrochemical performance of SESDs. Thus, leaching combined with rinsing is an effective strategy to decrease the impurities of biomass carbon. 4. Due to the limitations of synthesis equipment and technologies such as impurity cleaning and vacuum filtration, the continuous preparation of large-scale biomass electrodes is still a challenge worthy of attention. Therefore, the development of scale-up technology for preparation is an important issue as well. 5. An even more challenging aspect is the electrochemical shortcomings, such as the low initial Coulombic efficiency in SIBs, routine cycle performance, unsuitable voltage plateau, and poor energy density. Nanostructured strategies, such as structure/composition engineering, doping, and hybridization with active materials, are demonstrated to be the best potential choices to enhance the performance in SIBs and SICs. It is important to further elucidate the Na-ions storage mechanisms and better explore biomass-based materials with controllable microstructures.

Lastly, the existing studies of new biomaterial systems and synthesis strategies have provided a new platform for the development of SESDs, and a lot of work is still needed in the future.

## Figures and Tables

**Figure 1 nanomaterials-12-00930-f001:**
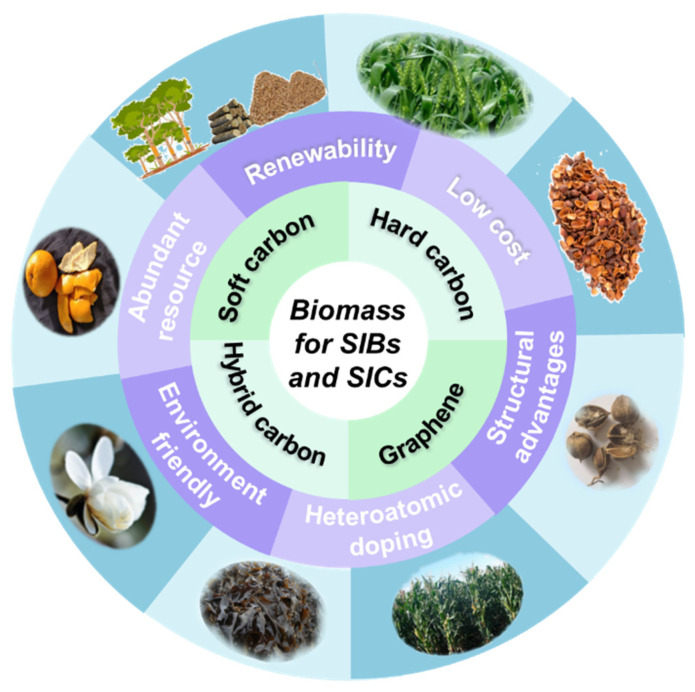
The schematic diagram of biomass-based electrode materials from different precursors, and their inherent advantages.

**Figure 2 nanomaterials-12-00930-f002:**
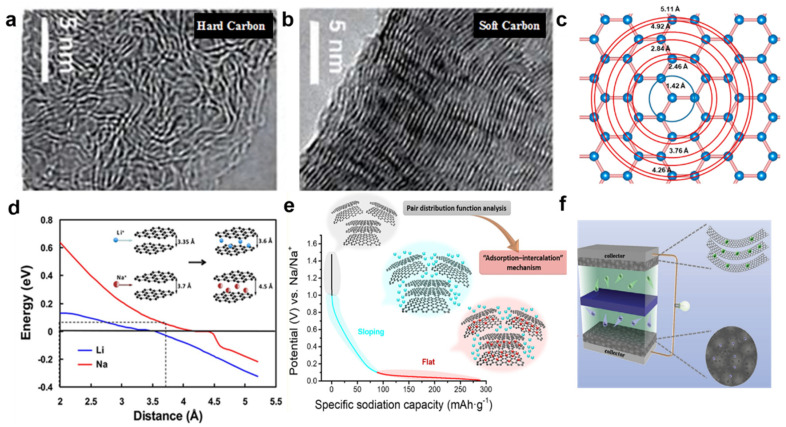
TEM image of hard carbon (**a**) and soft carbon (**b**). Reprinted with permission from Ref. [67]. Copyright 2017 American Chemical Society. (**c**) Schematic of the atomic structure of graphene and near-neighbor interatomic distances. Reprinted with permission from Ref. [68]. Copyright 2019 American Chemical Society. (**d**) Theoretical energy cost for Na (red curve) and Li (blue curve) ions insertion into carbon as a function of carbon interlayer distance. Reprinted with permission from Ref. [69]. Copyright 2012 American Chemical Society. (**e**) Schematic illustration of the mechanism for Na storage in hard carbon. Reprinted with permission from Ref. [68]. Copyright 2019 American Chemical Society. (**f**) Schematic of SIBs. Reprinted with permission from Ref. [70]. Copyright 2017 Wiley-VCH.

**Figure 3 nanomaterials-12-00930-f003:**
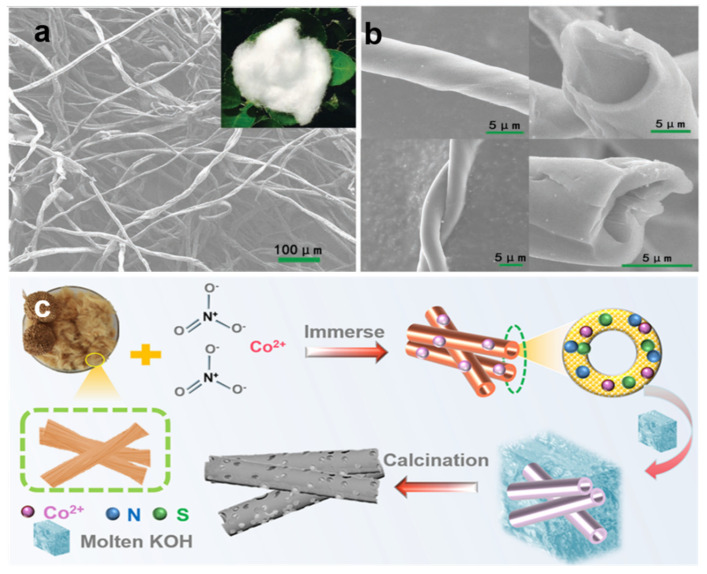
(**a**) SEM image and photograph of cotton. (**b**)The magnified SEM images of the carbonized cotton with the detailed structure information. Reprinted with permission from Ref. [46]. Copyright 2017 Wiley-VCH. (**c**) Illustration of the preparation process of the cross-linked porous sample. Reprinted with permission from Ref. [79]. Copyright 2021 American Chemical Society.

**Figure 4 nanomaterials-12-00930-f004:**
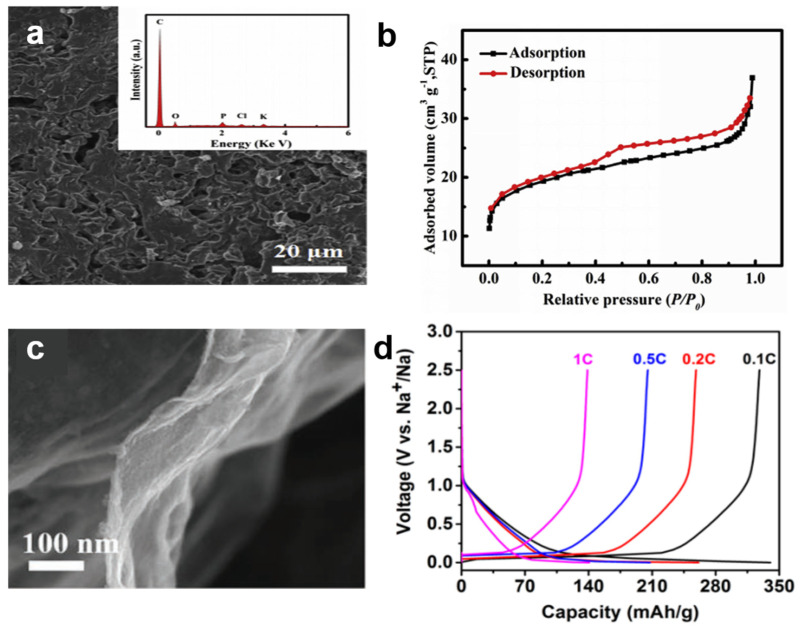
(**a**) Scanning electron microscopy (SEM) image and (**b**) N_2_ adsorption–desorption isothermal curves of hard-carbon nanosheets from the pyrolysis of oat flakes. Reprinted with permission from Ref. [82]. Copyright 2019 Elsevier. (**c**) SEM image of hard-carbon nanosheets from the pyrolysis of biomass-based gelatin. Reprinted with permission from Ref. [83]. Copyright 2020 Wiley-VCH and (**d**) galvanostatic charge/discharge cycling profiles of maple-derived hard carbon. Reprinted with permission from Ref. [84]. Copyright 2019 Elsevier. 2D nanostructured carbons, with their highly specific surface areas, continuous electron conduction paths, and ability to maintain volume changes during charge and discharge, have attracted much attention for Na-ion storage.

**Figure 5 nanomaterials-12-00930-f005:**
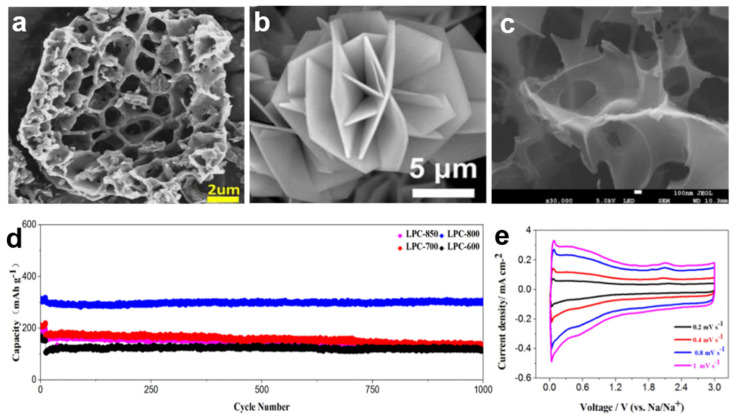
(**a**) SEM images of “honeycomb” structure carbon. (**b**) SEM images of CoP/C. Reprinted with permission from Ref. [89]. Copyright 2018 American Chemical Society. (**c**) SEM images of porous carbons from longan shells. (**d**) Cycling stability performance of different samples at a current density of 5 A g^−1^. (**e**) CV curves of porous carbons. Reprinted with permission from Ref. [88]. Copyright 2018 Elsevier.

**Figure 6 nanomaterials-12-00930-f006:**
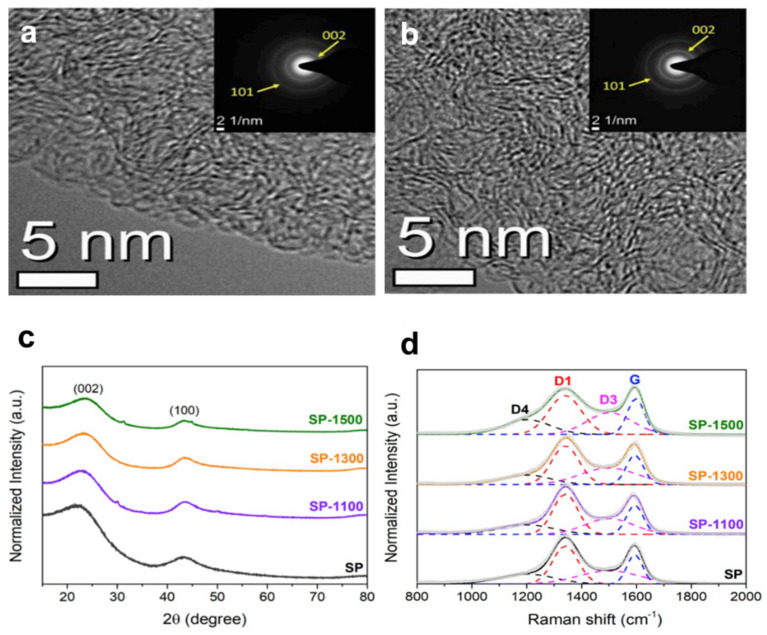
TEM images (**a**,**b**), XRD patterns (**c**) and Raman spectra (**d**) of hard carbons derived from fir wood. Reprinted with permission from Ref. [105]. Copyright 2019 Elsevier.

**Figure 7 nanomaterials-12-00930-f007:**
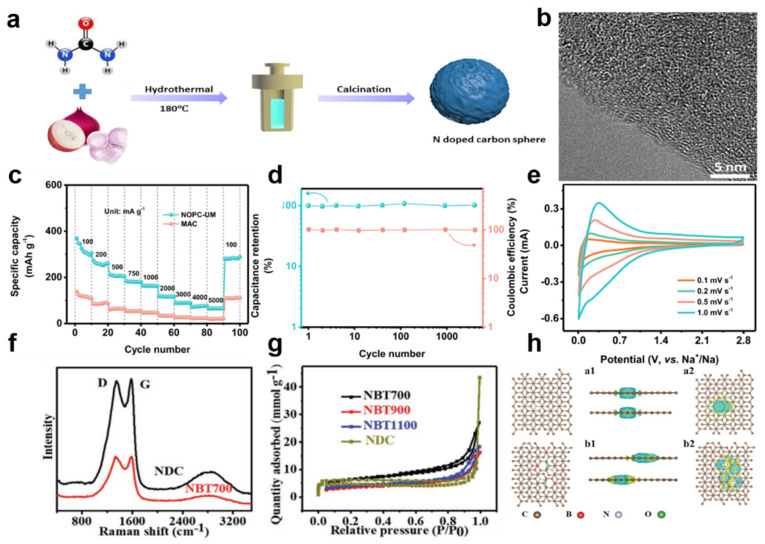
(**a**) Schematic illustration of the synthesis of N-doped carbon sphere. (**b**) TEM image of carbon sphere. Reprinted with permission from Ref. [111]. Copyright 2020 Elsevier. (**c**) The rate performance of NOPC and MAC anodes at various current densities. (**d**) Cycling stability of hard carbon anode measured at 1000 mA g^−1^. (**e**) CV curves measured of hard carbon anode. Reprinted with permission from Ref. [5]. Copyright 2020 Elsevier. (**f**) The Raman spectra of NDC and NBT. (**g**) The N_2_ adsorption–desorption isotherms at different temperature. (**h**) The N doping model for NDC and N, B co-doping model for NBT, respectively. Reprinted with permission from Ref. [83]. Copyright 2021 Wiley-VCH.

**Figure 8 nanomaterials-12-00930-f008:**
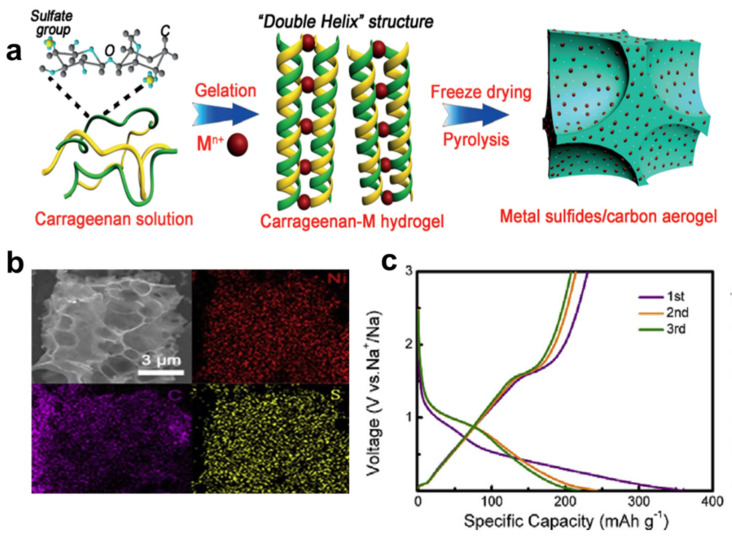
(**a**) Schematic illustration of double helix structure. Reprinted with permission from Ref. [117]. Copyright 2021 Wiley-VCH. (**b**) The EDS mappings of Ni, C, and S elements. (**c**) The Ni_3_S_4_/C cycled at the 1st, 2nd, and 3rd from 0.01 V to 3.0 V (vs. Na^+^/Na) at a current density of 1 A g^−1^. Reprinted with permission from Ref. [118]. Copyright 2019 Elcevier.

**Table 1 nanomaterials-12-00930-t001:** Physical and electrochemical properties of various carbons.

Classification	Precursor	Yield [%]	Morphology	Application	Capacity [mAh g^−1^] /Current density [mA g^−1^]	ICE ^[a]^ [%]	Refs.
Graphite	/	/	sheets	SIB	284/20	49.53	[36]
	/	/	spherical	SIC	221 ^[b]^/500	/	[37]
Graphene	Graphite	/	sponges	SIB	372.0/50	67.4	[38]
	Graphite	/	nanosheets	SIB	240/200	52	[39]
	Cellulose/chitosan/GO	/	Layers	SIB	395/100	/	[40]
	Graphite	/	folded texture	SIC	115.6/100	/	[41]
	Graphite	/	porous	SIC	420/100	/	[42]
Soft carbon	Coal	/	porous	SIB	267/500	34.0	[43]
	Pitch	70	porous	SIB	268.3/100	82	[44]
Hard Carbon	Kapok	<10	tube	SIB	290/30	80	[45]
	Cucumber stems	/	porous	SIB	337.9/50	64.9	[46]
	Cherry petals	/	nanosheets	SIB	310/20	67.3	[47]
	Pine pollen	/	porous	SIB	370/100	59.8	[48]
	Longan shell	/	porous	SIB	345/100	73	[49]
	Leonardite humic acid	60.73	flakes	SIB	345/100	73	[50]
	gelatin	/	nanosheets	SIB	309/200	84.1	[51]
	Mushroom stalk	/	porous	SIB	305/100	33.8	[5]
	Samara	/	porous	SIB	333.2/100	35.7	[52]
	Chlorella	/	nanoparticle	SIB	436/100	51	[53]
	Carrageenan	/	double-helix	SIB	380/100	56.3	[54]
	Enteromorpha	/	sponge	SIC	362/100	/	[55]
	Carboxymethyl cellulose	/	porous	SIC	322/50	/	[56]

^[a]^ ICE = initial Coulombic efficiency. ^[b]^ F g^−1^.

## Data Availability

Not applicable.

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
