# Peer review of "Recent Advances in Biomass-Derived Carbon Materials for Sodium-Ion Energy Storage Devices"

_nanomaterials, 2022, doi:10.3390/nano12060930_

Round 1

Reviewer 1 Report

Authors reviewed “Recent advances in biomass-derived carbon materials for sodium-ion energy storage devices”.

This work is framed well and summarized neatly. Authors gave detailed information about mechanism of Na-ion batteries, Na-ion capacitors, and their configurations. Further, authors reported the details of bio mass carbon materials from notable references which seem to be interesting. However, I found few contents are missing and furnished those below.

This article needs major revision before acceptance for publication.

Detailed comments:

1. (line 35), “better performance in some cases” -- Elaborate this section by highlighting the specific performance parameters by referring the cited works in the article.

2. (line 45) – “Though the similarity between sodium and lithium chemistry, some differences still exist between them” – This sentence can be rewritten better.

3. Pg 3 (line 111) & Pg 8 (line 305) – Information about graphene is less. As this is a review article, more data can be included.

4. Sec 3.2 (line 212) Fig. 4 caption is not clear, There are two types of carbon derived from two different  source,  It is not clear the N2 absorption-desorption isotherms shown in b) is for which carbon ?, similarly, caption d) cycling profile. Further,   I don’t see any relevant discussions related to Fig.4 (a b c and d). 

5. Pg 7 (line 250) – The sizes of macropores, mesopores, and micropores can be mentioned in accordance with the cited.

6. Pg 11 (line 402) – “The relatively low carbon yield of precursors” – data related to yield from the cited papers and other works to be cited and appended in the manuscript-This would be useful to choose the right source to scale up and technology development.

7. In the conclusions and outlook (Pg 11 line 405), authors mentioned that “the development of precursors with high carbon yield and scale-up technology for preparation have become an important research direction”  -- You may also provide perspective view of other performance parameters such as capacity stability, low voltage plateau and physical characteristics like porosity, crystallinity, particle shape and size etc.  This would help future tailoring and value add to this review article.

Author Response

Reply: Thanks for the constructive suggestion. We have revised the manuscript based on your valuable comments. The modifications are marked up using the “Track Changes” function in the revised manuscript.

Detailed comments:

  1. (line 35), “better performance in some cases” -- Elaboratethis section by highlighting the specific performance parameters by referring the cited works in the article.

Reply: We revised the sentence and cited some related works in revisions.

  1. (line 45) – “Though the similarity between sodium and lithium chemistry, some differences still exist between them” – This sentence can be rewritten better.

Reply: We rewrote this sentence in the revised manuscript.

  1. Pg 3 (line 111) & Pg 8 (line 305) – Information about graphene is less. As this is a review article, more data can be included.

Reply: We have added some discussion about graphene electrodes in SIBs and provided more electrochemical data by referring to some representative works. (line 120 and line 346 )

  1. Sec 3.2 (line 212) Fig. 4 caption is not clear, There are two types of carbon derived from two different source, It is not clear the Nabsorption-desorption isotherms shown in b) is for which carbon?, similarly, caption d) cycling profile. Further, I don’t see any relevant discussions related to Fig.4 (a b c and d). 

Reply: We rewrote the section about Figure 4 in Pg 7 and revised the caption.

  1. Pg 7 (line 250) – The sizes of macropores, mesopores, and micropores can be mentioned in accordance with the cited.

Reply: Based on your suggestion, we highlighted the size of different types of holes in Pg 9.  

  1. Pg 11 (line 402) – “The relatively low carbon yield of precursors” –data related to yield from the cited papers and other works to be cited and appended in the manuscript-This would be useful to choose the right source to scale up and technology development.

Reply: In previous works, researchers usually do not pay much attention to reports of carbon yield. However, it is widely accepted that the carbon yield of biomass is generally low. We have added some representative data on the carbon yield of different precursors in table 1.

  1. In the conclusions and outlook (Pg 11 line 405), authors mentioned that “the development of precursors with high carbon yield and scale-up technology for preparation have become an important research direction”  -- You may also provide perspective view of other performance parameters such as capacity stability, low voltage plateau and physical characteristics like porosity, crystallinity, particle shape and size etc.  This would help future tailoring and value add to this review article.

Reply: Thanks for your suggestion. We have added some perspectives on the research direction in this field based on our understanding.

Reviewer 2 Report

This bibliographic report is far too poor to be published as a review. Only a few selected papers are presented, without any systematic comparison of electrochemical performance according to morphology, structure and chemical composition. A very poor contribution for scientists wishing to have a quick overview of recent advances in the field. From this point of view, this article should be rejected for publication.

Author Response

Comments:

This bibliographic report is far too poor to be published as a review. Only a few selected papers are presented, without any systematic comparison of electrochemical performance according to morphology, structure and chemical composition. A very poor contribution for scientists wishing to have a quick overview of recent advances in the field. From this point of view, this article should be rejected for publication.

Reply: 

Thanks for the constructive suggestion. We have revised the manuscript based on your valuable comments. The modifications are marked up using the “Track Changes” function in the revised manuscript.

In fact, in this review, we have systematically summarized a lot of representative work based on biomass-derived carbons used for the sodium-ion energy storage system. We have described the principles and Na-ion storage mechanisms of carbonaceous materials in SIBs and SICs and reviewed the literature, with a focus on the influence of the structure, morphology, and composition of biomass-derived carbons on its electrochemical performance in the sodium-ion energy storage system. A comprehensive account of the recent progress and design principle of biomass-derived carbon as sodium-ion storage materials have been presented, and guidance in future rational tailoring of biomass-derived carbons also have been provided.

Reviewer 3 Report

Manuscript ID: NANOMATERIALS-1607021

Title: Recent advances in biomass-derived carbon materials for sodium-ion energy storage devices

Recommendation: Major revision required

Report

The review manuscript focuses on the sustainable bio-sources as an energy material for next-generation sodium-ion batteries. Recent progress and advancements met out with the above systems are summarized, however, the manuscript does not cover extensive discussions based on the biomass-based carbons. The overall study design seems to be acceptable, but the authors need to bring more discussion in multifarious sections as pointed out below in my comments. Considering the area of sustainable energy storage devices, which is one among the much-needed research area which this manuscript has undertook partially. Therefore, I recommend for a major revision as a significant revision has to be carried out. Below are my comments. 

  1. Hard to understand, suggest rearrange the sentence page 2 “In previous studies, a large variety of cathode materials (e.g., oxides and polyanionic compounds) have 50 been demonstrated can effectively store sodium ions. A bottleneck now shows up at the anode”
  2. There are several reviews based on biomass as biochar for energy storage devices. What is the key understanding that this manuscript brings about needs more clarity in the introduction part?
  3. Suggest elucidating the physical and electrochemical properties of carbons (hard, soft, amorphous, graphene, graphite, heteroatom doped carbons etc.,) in a table format to have depth understanding.
  4. The botanical name of biomass represented in the manuscript should be italics as per nomenclature. For instance, page 8 “Osmanthus fragrans” should be made italics.
  5. The structure “ear-washed spherical” is un-common, please remove from the conclusion part.
  6. Carbons based on biomass as energy sources are numerous, while each section in the manuscript represented only a single work which is unacceptable. Therefore, I strongly suggest adding more discussion based on several biomass-based carbon for sodium-ion storage.
  7. The storage mechanism of Na-ion battery as well Na-ion capacitors defers. What is the structure/morphology required to enhance the current performance of both devices needs to be emphasized?
  8. A section should be devoted such that future challenges be linked with the current improvements met out in Na-ion battery/capacitor.
  9. Few errors need to be fixed; some are provided below.

“Collage” in affiliation, “ear-shaped spherical” in conclusion

Author Response

Reply: Thanks for the constructive suggestion. We have revised the manuscript based on your valuable comments. The modifications are marked up using the “Track Changes” function in the revised manuscript.

1. Hard to understand, suggest rearrange the sentence page 2 “In previous studies, a large variety of cathode materials (e.g., oxides and polyanionic compounds) have 50 been demonstrated can effectively store sodium ions. A bottleneck now shows up at the anode”

Reply: we have revised this sentence to make it easier to understand in Pg 2.

2. There are several reviews based on biomass as biochar for energy storage devices. What is the key understanding that this manuscript brings about needs more clarity in the introduction part?

Reply: in this review, we attempt to provide a comprehensive summary of the latest developments of various biomass-derived carbons used in sodium-ion energy storage and the classification of biomass carbon with different structures and compositions. We revised the introduction part and highlighted the modification in red.

3. Suggest elucidating the physical and electrochemical properties of carbons (hard, soft, amorphous, graphene, graphite, heteroatom doped carbons etc.,) in a table format to have depth understanding.

Reply: We have made a summary of the properties of different types of carbons in table 1 based on your suggestion.

4. The botanical name of biomass represented in the manuscript should be italics as per nomenclature. For instance, page 8 “Osmanthus fragrans” should be made italics.

Reply: We have revised this question based on your suggestion.

5. The structure “ear-washed spherical” is un-common, please remove from the conclusion part.

Reply: we rewrote this section in Pg 9.

6. Carbons based on biomass as energy sources are numerous, while each section in the manuscript represented only a single work which is unacceptable. Therefore, I strongly suggest adding more discussion based on several biomass-based carbon for sodium-ion storage.

Reply: Based on your suggestion, we have revised this part of the different morphology of biomass-derived carbons used for Na-ion storage.

7. The storage mechanism of Na-ion battery as well Na-ion capacitors defers. What is the structure/morphology required to enhance the current performance of both devices needs to be emphasized?

Reply: The mechanisms of NIB and NIC are indeed different, so we discuss this issue in two sections. We have further revised the sections based on your suggestion.  

8. A section should be devoted such that future challenges be linked with the current improvements met out in Na-ion battery/capacitor.

Reply: we have added the forward-looking views of Na-ion storage fields based on your comments in Pg 13.

9. Few errors need to be fixed; some are provided below.

“Collage” in affiliation, “ear-shaped spherical” in conclusion

 Reply: Thanks for your kind reminder. We have corrected this error.

Round 2

Reviewer 1 Report

Authors reviewed “Recent advances in biomass-derived carbon materials for sodium-ion energy storage devices”.

 The revised version is better than the previous one. Conclusion section is satisfactory. This work may be accepted. However, there are few errors. This work needs MINOR revisions.

1) Page 9, (line 293), The word “cur-rent” should appear without hyphen.

2) There are two sets of Fig 5. Check and remove the old version.

3) Page 9, (line 281), There should be a space between an integer and its unit. Change “50nm” as “50 nm”

4) Authors have cited Ref. 78 for Figure 5a “honeycomb carbon”. The cited Ref. 78, doesn’t have any information regarding “honeycomb carbon”. The Ref. 78 contains information about “hollow carbon”. CROSS CHECK THE REFERENCES CITED throughout the manuscript.

Author Response

The revised version is better than the previous one. Conclusion section is satisfactory. This work may be accepted. However, there are few errors. This work needs MINOR revisions.

Thank you again for your kind comments. We have revised the errors you mentioned in the revisions and carefully checked it again.

Page 9, (line 293), The word “cur-rent” should appear without hyphen.

Reply: We corrected this mistake in revised manuscript (Pg 8).

2) There are two sets of Fig 5. Check and remove the old version.

Reply: We remove the old version in revised manuscript (Pg 8).

3) Page 9, (line 281), There should be a space between an integer and its unit. Change “50nm” as “50 nm”

Reply: We revised this case in Pg 8.

4) Authors have cited Ref. 78 for Figure 5a “honeycomb carbon”. The cited Ref. 78, doesn’t have any information regarding “honeycomb carbon”. The Ref. 78 contains information about “hollow carbon”. CROSS CHECK THE REFERENCES CITED throughout the manuscript.

Reply: We corrected this mistake. The information of “honeycomb carbon” is referred to Ref 82. Thanks for your kind reminder, we proofed all the references in the manuscript. The revised references are highlighted in yellow.

Reviewer 3 Report

Authors have revised manuscript as per the suggestions and I feel the quality of the revised manuscript certainly improved.

Author Response

Authors have revised manuscript as per the suggestions and I feel the quality of the revised manuscript certainly improved.

Thank you again for your kind comments.